# A tree-ring perspective on the past and future mass balance of a glacier in Tien Shan (Central Asia): an example from the Tuyuksu glacier, Kyrgyzstan

Youping Chen<sup>1</sup>, Magdalena Opała-Owczarek<sup>2</sup>, Feng Chen<sup>1, 3\*</sup>, Piotr Owczarek<sup>4</sup>, Heli

5 Zhang <sup>3, 1</sup>, Shijie Wang <sup>1</sup>, Mao Hu <sup>1</sup>, Rysbek Satylkanov <sup>5</sup>, Bakytbek Ermenbaev <sup>5</sup>, Bakhtiyorov Zulfiyor <sup>1, 6</sup>, Huaming Shang <sup>3</sup>, Ruibo Zhang <sup>3</sup>

<sup>1</sup>Yunnan Key Laboratory of International Rivers and Transboundary Eco-Security, Institute of International Rivers and Eco-Security, Yunnan University, Kunming, 650500, China

<sup>2</sup>Institute of Earth Sciences, Faculty of Natural Sciences, University of Silesia in Katowice, Ul. Bedzinska
 60, 41-200 Sosnowiec, Poland

<sup>3</sup>Key Laboratory of Tree-ring Physical and Chemical Research of the Chinese Meteorological Administration/Xinjiang Laboratory of Tree-ring Ecology, Institute of Desert Meteorology, Chinese Meteorological Administration, Urumqi, 830002, China

<sup>4</sup>Institute of Geography and Regional Development, University of Wroclaw, Pl. Uniwersytecki 1, 50-137
Wrocław, Poland

<sup>5</sup>Tien-Shan Mountain Scientific Center, Institute of Water Problems and Hydro Power, National Academy of Sciences of the Kyrgyz Republic, Bishkek, Kyrgyzstan;

<sup>6</sup>Khujand Science Center Academy of Sciences of the Republic of Tajikistan, Khujand, Tajikistan

Correspondence to: Feng Chen (feng653@163.com)

- 20 Abstract. The Tien Shan glaciers, known as "Central Asia's Water Tower," have a direct influence on water resource management in downstream parched areas. The limited time periods of currently available observational climate datasets hamper an accurate examination of glacial changes in Central Asia in terms of long-term climate change implications. In this work, we analysed this change by combining tree-ring-based reconstructions of the Tuyuksu Glacier's high-altitude mass balance during the last 382
- 25 years with models of the future mass balance of this glacier until the year 2100 CE. The results show that mountain precipitation is an important force driving the cycles of the cryosphere, biosphere and hydrosphere in arid Central Asia. This driving force has broad coherence in spatiotemporal variation, with periodic cycles and decadal shifts caused by the North Atlantic Oscillation and the El Niño-Southern Oscillation. The multi-model mean in CMIP6 suggests a downward trend in glacier mass balance until
- 2100, but this trend will be moderated by increased precipitation. The findings of the study could help to explain how the glacial mass balance has evolved in the Tien Shan Mountains of Central Asia throughout time and its relationship to other geosphere layers.

Keywords: Central Asia; Tuyuksu Glacier; mass balance reconstruction; Tree-ring; CMIP 6

#### **1** Introduction

- Numerous papers have reported rapid glacier melt worldwide resulting from ongoing global warming, the effects of which are particularly obvious in mountainous areas, especially in Central Asia, where temperatures have recently been at a high level (Xu et al., 2018; Zhang et al., 2019). Continuous glacier melt will have a significant impact on Central Asia's freshwater supplies, particularly in high-emission scenarios (Huss and Hock, 2018; Li et al., 2019; Li et al., 2020). The major Central Asian river basins of
- the Tarim, the Amu Darya and the Syr Darya will receive their maximum glacier meltwater input during the next few decades (Huss and Hock, 2018). This increases the likelihood of freshwater shortages and could lead to conflicts in fast-developing economies (Munia et al., 2016). The most direct evidence of a warming influence on glaciers is the glacier mass balance, which governs ice dynamics and glacier behaviour (Hagg et al., 2017; Azam et al., 2020; Liu et al., 2020; Shean et al., 2020). But because most
- glacier mass balance data from around the world are less than 50 years old, a comprehensive evaluation of glacier variations and responses to climate change on an inter-annual and decadal scale is impossible (Cerrato et al., 2020). Nevertheless, since glacier mass balance is largely influenced by climate, other proxies sensitive to the same factors can be used to reconstruct past mass balance series (Cerrato et al., 2020; Managave et al., 2020; Singh et al., 2021).
- Tree rings are a valuable and sensitive proxy for recording past changes in glacial mass balance. In Asia, Singh (2021) employed a tree-ring isotope chronology from mixed conifers to reconstruct the fluctuation in glacial mass balance in the central Himalayas from 1743, finding that the loss of mass balance had accelerated since the 1960s. In Europe, Cerrato (2020) likewise employed tree-ring densities of Swiss stone pines to recreate the Careser Glacier's summer mass balance fluctuation. In previous
- research in Central Asia, tree rings were successfully applied to reconstruct precipitation, temperature and runoff variations in mountainous locations, but they were rarely employed to reconstruct glacial mass balance. Following the demise of the Soviet Union, a vast number of observations of huge glacier mass balances were halted. There have only ever been two long-term mass balance programmes: one at the Urumqi Glacier No. 1 in China and the other at the Tuyuksu Glacier in Kyrgyzstan. Such long-term
- observations are a good basis for understanding past changes in glacier mass balance based on tree rings.

Few studies have hitherto utilized tree rings to analyse the Tuyuksu Glacier's mass balance, although it is worth noting that Zhang et al. (2019) employed annual ring isotopic chronology to estimate the Tuyuksu Glacier's mass balance variations from 1849 to 2014, finding that after 1968, melting was taking place without interruption and faster than at any time before.

- We can achieve a better grasp of the variations in the Tuyuksu Glacier's mass balance on the basis 65 of longer and more comprehensive data. In this study, we have sought (1) to explore the interplay between 66 tree rings (biosphere) and glacial mass balance (cryosphere), precipitation and runoff (hydrosphere); (2) 76 to undertake a reconstruction of the Tuyuksu Glacier's high-altitude mass balance variations between 76 1635 and 2016 CE based on tree-ring and mass balance responses, and to contrast the spatial and temporal
- characteristics of our reconstruction with other glacier balance masses and runoffs based on tree ring reconstruction; and (3) to look into the relationship with large-scale climatic forcings and predict how this will change in the future. The findings of this study will advance our knowledge of how climate change affects the mass balance of the Tuyuksu Glacier, as well as changes in tree growth and runoff around it, in the context of long-term scales.

## 75 2 Materials and methods

## 2.1 Geographical Settings

Figure 1: Distribution map of Tuyuksu Glacier and its surrounding main rivers in the Tien Shan Mountains of Central Asia (a). Location of tree-ring sampling sites, Karaoy hydrological station and CRU meteorological grid dataset around Tuyuksu Glacier (b).

Spanning most of Central Asia, with numerous glacial fluctuation sequences, the Tien Shan Mountains