# Peer review of "A tree-ring perspective on the past and future mass balance of a glacier in Tien Shan (Central Asia): an example from the Tuyuksu glacier, Kyrgyzstan"

_Hydrology and Earth System Sciences, 2022_

## Author Comment (AC1)

Dear editors and reviewers:

On behalf of my co-authors, we thank you very much for giving us an opportunity to revise and modify our manuscript, we appreciate you and reviewers for their constructive comments and suggestions on our manuscript entitled "A tree-ring perspective on the past and future mass balance of a glacier in Tien Shan (Central Asia): an example from the Tuyuksu glacier, Kyrgyzstan" (Ms. No. hess-2022-329).

We have studied reviewer's comments carefully and have made revision and modification. Please find below a detailed responses to all the raised points, which we would like to submit for your kind consideration.

We would like to express our great appreciation to you and reviewers for comments on our paper. Looking forward to hearing from you.

Thank you and best regards.

Yours sincerely,

Feng Chen

On behalf of all authors

**Reviewer #1:** Specific comments**

L 32: 'geosphere layers' is not clear, please clarify the meaning

Response: Yes, you are right, we have corrected it to "biosphere and hydrosphere".

L 45: what is the meaning of 'comprehensive'? Worldwide? Century-long? Others? Please clarify **Response:** The "comprehensive" here mainly refers to the long time scales of interannual and interdecadal periods. We have changed the loose expression "comprehensive" to "long-term".

L 52: finding that the loss of mass (remove 'balance')

Response: Yes, we have removed "balance".

L 57: huge = large glaciers? Or maybe long series of mass balance? Please clarify

Response: We have removed the incorrect expression "huge".

L 58: Two long-term mass balance programs exist (where?): one at the Urumqi...... It is not clear if these series are ongoing or have been demised

**Response:** We supplemented the longitude and latitude information (86.82° E, 43.08° N) of Urumqi No. 1 Glacier., and the observation sequence of this glacier is still being updated.

L 60: ... for reconstructing past changes

Response: Yes, we have corrected

L61: The mass balance of the Tuyuksu Glacier was analysed by Zhang et al. (2019), who employed..... (remove the first sentence)

Response: Yes, you are right, we have corrected it.

L 69: and to compare the spatial and temporal characteristics of our reconstruction with other glacier mass balance and runoff reconstructions based on tree-ring

Response: Thank you for your suggestion, we have corrected it as requested.

L71: predictions are always hard to do, especially of the future! Please replace with 'project', or 'try to understand the possible future evolution'

**Response:** Yes, the description here is not rigorous, thank you for your suggestion, we have corrected it to "try to understand the possible future evolution".

L 74: long-term time scales? Long -term variations?

Response: We have corrected to "long -term variations".

L 89: coring sites are 100 km west of the glacier, not really 'around' it

**Response:** Yes, you are right, we have corrected the misstatement there.

L 90: human 'agencies'? Please clarify

Response: We corrected it to "human activities".

L 107: maybe better 'Based on the threshold values, the RC starts in 1635 CE (figure 2).'

**Response:** Thank you for your suggestion, we have corrected it to read "the RC starts in 1635 CE".

L 111: we retrieved annual balance data from the WGMS database..... The data are average annual values at 100 m elevation bands comprised between 3400 and 4200 m, and cover the period between 1969 and 2016.

Response: Thank you for your suggestion, we have corrected it according to your request.

L 133: I do not understand the meaning of the sentence 'Calculations show that fluctuations in the glacier mass balance at various elevations are quite constant'

**Response:** Sorry for our incomprehensible expression here. What we want to express here is: the correlation analysis shows that the change trend of glacier mass balance at different altitudes is very consistent during 1969-2016.

L 115: what are 'simulation parameters'? Simulated/reconstructed (or maybe measured?) meteorological variables? Air temperature and precipitation? In addition, it is unclear whether the authors simulated the mass balance of the glacier in the past/future with these meteorological data, or maybe they only did statistical analyses...

**Response:** Sorry, we didn't express as well as we wanted here. "Simulation parameters" refer to simulated values of the mass balance of the glacier. This simulated value is based on the mean temperature and precipitation ensemble means of the 3 models (CanESM, CESM2, CESM2-WACCM, 33 ensemble members) of Phase 6 of the Coupled Model Comparison Project and scaled by a linear model, which were built from instrumental glacier mass balance and instrumental temperature and precipitation. We have made corresponding modifications in

this section.

L 125 and following: please make more explicit which period these statistics are referred to **Response:** Thanks for pointing out the deficiencies, we have added the period "1969-2016".

L 128: please do not mix annual with monthly data.

Response: Yes, we have corrected it.

L 137-140: this part is poorly written, please rephrase. In addition, why Pearson and not e.g. Spearman correlation analyses? Are the frequency distributions of analysed variables normal? Are the relationships linear and outliers absent? And what about the sample size? Please check e.g. https://doi.org/10.2307/2346598 and justify the choice of Pearson vs. Spearman correlations.

**Response:** Sorry for the poor writing here. We have rewritten it. Instead, we choose Pearson analysis because it is mainly used to evaluate the linear relationship between two variables. And in other studies on the correlation between tree-ring and climate and hydrology, Pearson analysis is basically used.

What is the 'altitude combination of glacier mass balance'? Is it the mass balance averaged for different elevation bands?

Response: Yes, as you understand, it is the average mass balance of different altitude bands.

L 141: bases on the correlation analysis in the period between (1969 to 2016 I guess...), a linear....employed in the period between (1635 and 1969?) to reconstruct.... mass balance based on tree-ring (and/or meteorological data?). Hard to follow here...

**Response:** We apologize for being unclear here. As you said, we first analyzed the significant positive correlation between tree-ring and glacier mass balance from 1969 to 2016, and then further based on this significant correlation, used a linear model to reconstruct the glacier mass balance sequence from 1635 to 2016. We have supplemented the corresponding period in the manuscript.

L 145: why 31-year and not other lengths? Following period requires rewriting because it is poorly written.

**Response:** Our choice to use 31-year low-pass values is based on reference to several similar papers that reconstruct past climate change based on tree-ring (eg. Fowler et al., 2012). At the

same time, the comparison shows that the results of using 31-year low-pass values or using 21year low-pass values are similar.

L 152: why is Oscillation Period uppercase?

**Response:** We have corrected it to lowercase.

L 153: why 'meanwhile'? Please remove it.

Response: Yes, we have removed it.

L 160-167: it would be interesting to know whether the maximum correlation matches with the zerobalance ELA or is just above it, as it looks like in Fig. 4.8.4 page 77 in: WGMS 2021. Global Glacier Change Bulletin No. 4 (2018–2019). Zemp, M., Nussbaumer, S.U., Gärtner- Roer, I., Bannwart, J., Paul, F., and Hoelzle, M. (eds.), ISC(WDS)/IUGG(IACS)/UNEP/UNESCO/WMO, World Glacier Monitoring Service, Zurich, Switzerland, 278 pp., publication based on database version: doi:10.5904/wgms-fog-2021-05.

**Response:** Yes, the idea is novel. Correlation analysis showed that there was a significant negative correlation between ELA and tree-ring index from 1969 to 2016 (r = -0.361, p < 0.05). We further analyzed the correlation between annual precipitation and temperature and ELA, the results showed that ELA was significantly positively correlated with the average temperature from April to October (r=0.473, p < 0.01), and significantly negatively correlated with the average with the average precipitation from July to September (r = -0.584, p < 0.01).

In the following, the authors work mainly with mass balance data above 3800 m, which mean they reconstruct the mass balance in the accumulation area of the glacier. This should be state more explicitly in the manuscript.

**Response:** Yes, thanks for your suggestion. We have made this clear in the corresponding section of the manuscript.

L 168: aided -> positively correlated?

**Response:** We have corrected to "Both tree radial growth and glacier mass balance can be affected by precipitation".

L 176: I suggest being less categorical, e.g. 'Overall, our findings indicate that fluctuations in the Tuyuksu Glacier's high-altitude mass balance are influenced by meteorological conditions during spring and summer. In particular, precipitation during spring and summer looks comparatively more important than air temperature in the same period of the year.

**Response:** Thanks for your suggestion, we have corrected the imprecise statement here.

L 184: it is unclear from the text and the figure whether the correlation for precipitation is highest for the period going from the previous july to the current june (as I understand). Please report always the same number of decimals (2 or 3), both in figures and text.

**Response:** Thanks for your suggestion, we have modified the number of decimal points for the highest correlation coefficient in the text to 3 digits, consistent with the figure.

L 186: Figure 4 presents also correlations with runoff, which are not reported in the text. In my opinion it should be also interesting to se the relationship between precipitation and temperature, because from the Figure 4 it looks like they are negatively correlated, i.e. high runoff corresponds with low temperature and high precipitation in spring and summer. This could suggest that anticyclonic weather is associated with high temperature and low precipitation, and vice versa....

**Response:** Yes, you are right. Regarding the runoff-climate response, we place it in the third paragraph of the Discussion (4.1). In the first paragraph of the Results section (3.1), we first detected a significant positive correlation between glacier mass balance and tree-ring. Precipitation is a common influencing factor of glacier mass balance and tree-ring. Therefore, logically, we further explored the response relationship between glacier mass balance and tree-ring and precipitation respectively, and inspired the following reconstruction of glacier mass balance based on tree-ring. In the Discussion section, we first demonstrate that it is reasonable to reconstruct glacier mass balance based on tree-ring, because precipitation is a common driver of tree-ring and glacier mass balance. Then, based on the significant positive correlation between runoff and precipitation, we further conclude that precipitation is the main driving factor of tree-ring, glacier mass balance and runoff, that is, precipitation is the main driving factor of biosphere, cryosphere and hydrosphere. Therefore, for the sake of logic, we put the response relationship between runoff and precipitation in the Discussion section.

L200: please specify that these statistics are related to the leave-one-out validation. These statistics are ok, but I would suggest using RMSE (expressed in mm or m water equivalent) and/or the Nash and Sutcliffe index, which are more commonly used in the scientific literature (and therefore ease

comparisons with it).

**Response:** Yes, thanks for your suggestion, we have included a description of leave-one-out validation and model validation in the Methods section of the manuscript. In addition, we also added the RMSE parameter, which is 333.999.

L 218: a significant 'mass loss' trend during... Again, the authors should stress that they are dealing with the accumulation area of the glacier, which is only half of the story (the glacier had prevailing negative mass balance since the late 1970s, WGMS (2021)).

**Response:** Yes, thanks for your suggestion, our description here is not rigorous, and the description of the glacier accumulation area has been added here.

L 221: what is MTM? Please introduce acronyms

Response: Yes, we have corrected it to "Multi-Taper Method".

L 226: remove 'highly'

Response: Yes, we have corrected it.

L 229: correspond to SST (introduce acronym) anomalies linked to NAO and ENSO occurrences. Please state more explicitly what is the relationship and what are the SST anomalies linked to positive/negative ENSO and NAO phases (for the non-specialist readers).

**Response:** We added the full name of SST "sea surface temperature". In addition, we supplemented the definitions of NAO and ENSO and the relationship between their positive and negative phases and SST in the "2.3 Glacier mass balance and meteorological data" section of the manuscript. The content is as follows: "Among them, NAO is the inverse relationship between the Icelandic low pressure and the Azores high pressure over the North Atlantic Ocean, and its interdecadal atmospheric variability has a good consistency with the North Atlantic sea surface temperature (Deser and Blackmon, 1993). ENSO is an oscillation of wind field and sea surface temperature that occurs in the equatorial eastern Pacific region. El Niño in ENSO refers to the sea temperature in the central and eastern Pacific region, while Southern Oscillation refers to the sea temperature in the Western Pacific-Indian Ocean region. They are anti-correlated (Bamston et al., 1997).".

L 241: what are the 'regional climatic inputs'? Please explain

**Response:** Sorry for the misexpression here. We have corrected to "a composite of climate inputs".

L 244: I suggest removing the sentence 'which contributes to the build-up of the mass balance of high-altitude glaciers'

**Response:** Yes, you are correct, we have removed this sentence.

L 246-247: poorly written and hard to understand, please rephrase.

**Response:** We apologize for the lack of clarity here. We have corrected it to "A similar highaltitude glacier mass balance-precipitation response has also been reported in the Tien Shan Mountains (Yang et al., 2021)".

L 250-267: I am a non-specialist of these arguments and had to read three times this part to grasp what the authors mean. This means that, probably, the authors should clarify this part of the paper. Moreover, I suggest a more consistent use of verb tense, probably present is more suitable to describe the growth behaviour of analysed tree species.

**Response:** Sorry for the poor expression here, we have corrected it as follows: "On the other hand, the radial growth of *Picea schrenkiana* and *Juniperus turkestanica* is also mainly affected by precipitation during the pre-growing and growing seasons. Physiological explanations are as follows: First, July–September of the previous year was the middle and late period of tree radial growth, accompanied by an average temperature of 13.8°C and a total precipitation of 62.0 mm (Fig. 3b). Such hydrothermal conditions are conducive to trees growing larger leaves, accumulating photosynthetic substances, and further benefiting photosynthesis and precipitation absorption in the next year (Fritts, 1976). Secondly, previous October-April is the non-growing season for trees, accompanied by an average temperature of -5.3°C and a total precipitation of 193.7mm (Fig. 3b). Adequate snowfall during this period allows the trees to absorb more water during the early growing season of current year (Díaz et al., 2002). Finally, May-June of current year is the critical season for radial tree growth, accompanied by an average temperature of 10.9°C and a total precipitation of 101.9 mm (Fig. 3b). Increased rainfall during this period is conducive to increasing soil moisture and further producing cambium cells (Liu et al., 2011).".

L 269: this is the first time such a high correlation coefficient is presented. Among which variables is it calculated? Not clear

**Response:** We apologize for not being clear here. The correlation here represents to the correlation between the total precipitation from April to September and the total precipitation from previous July to June during 1969-2016.

L 282: I agree with these considerations but I wonder why there is a significant positive correlation with temperature in the winter months. It would be interesting to read what the authors think (or better, propose as possible explanation) about that.

**Response:** This may be due to higher altitudes and cooler climates where cooler winter temperatures delay snowmelt and thus store sufficient moisture for the growing season. Such responsive relationships also appear in other parts of Central Asia. For example, the tree-ring chronology in northern Kyrgyzstan developed by Zhang et al. (2020) is also positively correlated with the temperature from previous November to March.

L 286-290: poorly written, please rewrite e.g. 'Since precipitation has a strong link with RC and GMB3800-4100, we investigated the spatial correlation between GMB3800-4100 (RC is not presented....) and gridded precipitation from 1969 to 2016. A large geographic area showing significant correlation up to (up to....?) was found, encompassing eastern Kyrgyzstan and south-eastern Kazakhstan (figure 8).'

**Response:** Thanks for your suggested correction. We apologize for the poor writing here and have corrected it as you suggested.

L294: our reconstruction of what?

Response: Yes, we have corrected it to "Reconstruction of mass balance of Tuyuksu Glacier".

L 296: were matched to a high degree -> correlates well (or are highly correlated)

Response: Yes, you are correct, we have corrected it.

L 298: were relatively negative -> were characterised by negative mass balance.

Response: Yes, you are correct, we have corrected it.

L 299: confirm the robustness of our model approach (is this the intended meaning?)

Response: Yes, you are correct, we have corrected it.

L 302: I suggest writing explicitly that the paper presents results for the accumulation area of the glacier. Is it the same in the paper of Zhang et al., (2019)? Measurements clearly show that the mass balance of the glacier, in its entirety, was negative in the last decades (WGMS, 2021) and these considerations are important while discussing the results. I suggest linking better the sentence 'We extended the reconstruction of this glacier's mass balance sequence back to 1635, which is 216 years longer (than what?)'. E.g. 'even if our reconstruction regards only the accumulation area of Tuyuksu Glacier, the added value of our work is that we reconstructed mass balance in a longer period etc...'

**Response:** Thank you for your pertinent suggestion. You are correct that the overall mass balance of glaciers has been negative over the past few decades, including the study by Zhang et al. (2019), which also showed that the mass balance of glaciers has shown a melting trend since 1968. But as you said, the study by Zhang et al. (2019) is based on the entire glacier (3400-4200 altitude), while our study area is the accumulation area above the glacier (3800-4100 altitude). Therefore, we correct the discussion to "The reason may be that the mass balance sequence we reconstructed is for the accumulation area above the glacier, where the cold air temperature and more snowfall offset the melting effect of temperature on the ice and snow, while Zhang et al. reconstructed the mass balance of the entire glacier. However, it is worth noting that even though our reconstruction only considers the accumulation zone of the Tuyuksu Glacier, the added value of our work is that we reconstruct the mass balance over a longer period".

L 304: compared is actually better than 'contrasted' (in my opinion)

Response: Yes, we used "compared".

L 312: our glacier reconstruction sequence were negative -> our glacier reconstruction show negative mass balance between 1760 and 1779 (this is an example; I suggest rephrasing accordingly, or similarly, the rest of the paper). Sequence should be replaced with 'time series' in my opinion

**Response:** Yes, thank you for your correction, we have corrected it according to your comments, including other similar places in the full text.

L 322: here the authors present the periods when their model agree with other reconstructions. Are there only agreement period or also disagreement periods? The second ones are interesting as well!

**Response:** Only the agreement period is included here.

L 328: How were they scaled to zero?

**Response:** Sorry for the inaccurate expression here. We calculated the z-score for each series based on its mean and standard deviation.

L 335: in Figure 9 there are also periods with disagreement, see comment L 322

Response: Yes, we only emphasized agreement period in these comparisons.

L 350: it would be nice to see a figure depicting mean geopotential height anomalies (and dominant winds/storm tracks) associated to the positive/negative NAO phases in this geographic area (or northern hemisphere), and positive/negative ENSO as well

**Response:** Yes, thanks for your suggestion, we have added corresponding water vapor flux composite maps in Figure 6. The corrected figure as follow:

---

## Author Comment (AC2)

Dear editors and reviewers:

On behalf of my co-authors, we thank you very much for giving us an opportunity to revise and modify our manuscript, we appreciate you and reviewers for their constructive comments and suggestions on our manuscript entitled "A tree-ring perspective on the past and future mass balance of a glacier in Tien Shan (Central Asia): an example from the Tuyuksu glacier, Kyrgyzstan" (Ms. No. hess-2022-329).

We have studied reviewer's comments carefully and have made revision and modification. Please find below a detailed responses to all the raised points, which we would like to submit for your kind consideration.

We would like to express our great appreciation to you and reviewers for comments on our paper. Looking forward to hearing from you.

Thank you and best regards.

Yours sincerely,

Feng Chen

On behalf of all authors

**Reviewer #2**:: The authors presented interesting results for their study glacier. However, from my taste, novelty of the paper is far from acceptable in this journal. Seems the authors showed lots of trust on correlation analysis. They used correlation coefficient to evaluate their models, to explore the relevance between two variables, and to compare with other studies. I would say, correlation coefficient is useful for examining the change pace of two variables but can't tell the bias. Moreover, the authors used only simple linear models to reconstruct and predict the glacier mass balance, which I think is of highly uncertain. First, the training of linear models is heavily related to the collected data. There area inevitable uncertainty in the data collection, especially when only data from one glacier was collected in this study. Second, validation of the linear models is poor in this study, especially for the prediction models. Although the authors claimed that they validated the reconstruction model by a leave-one-out method, there were few details on how this was conducted in the paper. Last, the linear models typically ignored other contributing factors beyond the considered ones. Glacier mass balance is typically controlled by many factors, such as ice flowing, radiation, albedo, and even terrain, not simply by precipitation and temperature. Prediction of future glacier mass balance based on solely precipitation or temperature is highly uncertain.

**Response:** Yes, a large number of research reveals the relationship between glacier mass balance and climate change. Most studies are based on qualitative or semi-quantitative analysis. Although these studies have built a complete story framework (I also admit that our story is not very novel). But our study is the first to show high-resolution changes in the upper accumulation area of the Tuyuksu Glacier over the past 382 years and possible future changes. As you pointed out, we show interesting results from studying glaciers. In Central Asia and even globally, there are very few research results that combine dendrochronology with glaciers, but are more used to study changes in precipitation, temperature, and runoff, as we described in the introduction of the manuscript.

In constructing linear models of past glacier mass balance. We admit that there are some uncertainties in the constructed linear model of glacier mass balance. However, in previous studies based on tree-ring in Central Asia, linear models have been widely and successfully used to reconstruct past climate and hydrological changes, and many interesting research results have been obtained (Zhang et al.2015; Chen et al., 2017; Panyushkina et al., 2018). In our study, tree-ring were significantly positively correlated with precipitation from previous

July to June ($r$= 0.665, $p$ < 0.01, 1969-2016). Referring to previous studies (Fritts, 1976; Zhang et al.2015), such a high correlation ($r$ > 0.6) allows tree-ring to be used to reconstruct past precipitation changes. In addition, the mass balance of high-altitude glaciers is also significantly positively correlated with precipitation from April to September ($r$= 0.675, $p$ < 0.01, 1969-2016). Therefore, on the basis that both tree-ring and high-altitude glacier mass balance are significantly related to precipitation, it is reasonable for us to use tree-ring to reconstruct glacier mass balance. In a similar way, Zhang et al. (2019) successfully reconstructed the changes of Tuyuksu Glacier based on tree-ring. On the other hand, as you said, after the linear model was established, we successfully used the leave-one-out method to verify the reliability of the model. As this method is a well-established method for verifying model reliability in dendrochronology (Michaelsen, 1987), we apologize for not detailing the process in the manuscript. In addition, we also acknowledge that there are uncertainties in estimating future changes in glaciers based on model simulation data from the Coupled Model Intercomparison Project 6 (CMIP6). However, previous studies have shown that the three models we selected can simulate climate change in Central Asia better than other models in CMIP6 (Guo et al., 2021). At the same time, we also use the method of multi-member ensemble averaging to eliminate the uncertainty of random coupling (33 ensemble members; Krishnamurti et al., 1999; Palmer al., 2000). In addition, previous similar studies have shown that data uncertainties in model simulations arise from internal variability in the climate system (Hessl et al., 2018 and Rao et al., 2020). On the other hand, there are relatively few research results on the combination of tree-ring data and model simulation data in the CMIP6. Our research is a useful attempt in this regard. In summary, although the linear model we constructed has certain uncertainties, it does not affect our understanding of past and future changes in the mass balance of the Tuyukesu Glacier.

Finally, we also acknowledge that, as you say, glacier mass balance is typically controlled by many factors, such as ice flowing, radiation, albedo, and even terrain, not simply by precipitation and temperature. However, it is undeniable that temperature and precipitation are the most important factors affecting the mass balance of glaciers (Sagredo et al., 2012; Cerrato et al., 2020). Low temperature and humid conditions are conducive to the accumulation of glaciers, while high temperature and drought conditions are conducive to the melting of

glaciers (Zhang et al., 2019). In other similar study, Cerrato et al. (2020) further used the tree-ring density to reconstruct the summer mass balance of the glacier based on the relationship between the tree ring density and the summer mass balance of the Careser Glacier, which were significantly correlated with the temperature from May to September.

By the way, I'm not familiar tree-ring. So I have a question for the authors, how did they have the tree-ring data for 1600s. Are there exactly trees so old in Central Asia? Regardingly, I would suggest the authors to provide more data (such as pictures) for the tree-ring collection. The simple statement of 'Data set available on request to corresponding authors' is unacceptable recently in HESS.

**Response:** Yes, there are many old trees in Central Asia. For example: Chen et al. (2022) collected tree-ring cores of *Larix sibirica* for nearly a millennial year in the Altay Mountains, and reconstructed the June-July temperature. Wang et al. (2021) collected tree-ring cores of *Picea schrenkiana* in the Tien Shan Mountains, and reconstructed the PDSI changes in the region over the past nearly 400 years. Davi et al. (2015) use an extensive collection of living and subfossil wood samples from temperature-sensitive trees to produce a millennial length, validated reconstruction of summer temperatures for Mongolia and Central Asia from 931 to 2005 CE. The following are some photos of our team collecting tree-ring cores in Central Asia:

[Figure]

**Some small comments:**

1. Seems the collection sites of tree-ring are far from the glacier. Do the authors have any comment on the representativeness of the tree-ring data?

**Response:** Yes, as you said, these sampling points are about 200 km away from the glacier. But as shown in Figure 8, we can see that the mass balance of glaciers and tree-ring is mainly affected by precipitation, and they are in the same high-correlation range. Therefore, we believe

that, using precipitation as a bridge, the tree-ring can represent changes in the mass balance of the accumulation zone above the glacier.

2. What is sample depth in Figure 2?

**Response:** "sample depth" refers to the number of samples, that is, the amount of tree sample cores collected.

3. Gridded data was derived from CRUs at a resolution of 0.5 degree, which is larger than the glaicer size. Do the authors have any comment of this uncertainty?

**Response:** Yes, it is undeniable that the accuracy of CRU grid meteorological data needs to be improved. However, as shown in the spatial correlation analysis of Fig. 8b, the gridded precipitation from April to September was significantly positively correlated with the glacier mass balance ($r > 0.5$), and this high correlation range included the entire glacier. Therefore, we think it is reasonable to use gridded CRU data for analysis.

4. Figure 3 different gradients should be different altitudes.

**Response:** Yes, thanks for the suggestion, we have corrected 'gradients' to 'elevations', including the same mistake in the title of Figure 4.

5. Leave more space between sub-panels in figure 4

**Response:** Yes, we have adjusted. The adjusted figure is as follows:

[Figure]

**Figure 4: Correlations of regional chronology with annual mass balance of the Tuyuksu Glacier at different elevations (a). Correlations between monthly precipitation and mean temperature from the CRU TS4.05 with the glacial annual mass balance at 3800-4100 m a.s.l (b), regional chronology (c) and annual runoff of Talas River (d). "P", "C" and "**" represent previous year, current year and 99% significant correlation, respectively**

6. What is growing season? From... to...

> **Response:** The optimum photosynthetic temperature for evergreen conifers ranges from 10 to 25 °C and that photosynthesis may cease at temperatures below -5 to -3°C or above 35 to 42 °C (Zhang et al., 2020). Figure 3b shows that the mean temperature in April and October was 4.7 °C and 3.7 °C. Thus, the period from April to October was regarded as the growth season for spruce trees in the study area.

7. Figure 8 is unclear or even unnecessary, I think . Are they discussing the spatial representativeness of the glacier or the precipitation data?

> **Response:** Such an analysis is necessary. As you suggested in points 1 and 3 above, such an analysis helps us understand how extensively precipitation drives changes in both tree-rings and the mass balance of the Tuyuksu Glacier.

**References**

Cerrato R, Salvatore M C, Gunnarson B E, et al. *Pinus cembra L.* tree-ring data as a proxy for summer mass-balance variability of the Careser Glacier (Italian Rhaetian Alps)[J]. Journal of Glaciology, 2020, 66(259): 714-726.

Chen F, Chen Y, Davi N, et al. Summer Temperature Reconstruction for the Source Area of the Northern Asian Great River Basins, Northern Mongolian Plateau Since 1190 CE and its Linkage With Inner Asian Historical Societal Changes[J]. Earth Science Frontiers, 2022, 10: 904851.

Chen F, Yuan Y, Yu S. Tree-ring indicators of rainfall and streamflow for the Ili-Balkhash Basin, Central Asia since CE 1560[J]. Palaeogeography, Palaeoclimatology, Palaeoecology, 2017, 482: 48-56.

Davi N K, Jacoby G C, Curtis A E, et al. Extension of drought records for central Asia using tree rings: West-central Mongolia[J]. Journal of Climate, 2006, 19(2): 288-299.

Fritts, H.: Tree rings and climate, Academic Press, London, 1976.

Michaelsen J. Cross-validation in statistical climate forecast models[J]. Journal of Applied

Meteorology and Climatology, 1987, 26(11): 1589-1600.

Panyushkina I P, Meko D M, Macklin M G, et al. Runoff variations in Lake Balkhash Basin, Central Asia, 1779–2015, inferred from tree rings[J]. Climate Dynamics, 2018, 51(7): 3161-3177.

Sagredo E A, Lowell T V. Climatology of Andean glaciers: A framework to understand glacier response to climate change[J]. Global and Planetary Change, 2012, 86: 101-109.

Wang T, Bao A, Xu W, et al. Tree-ring-based assessments of drought variability during the past 400 years in the Tianshan mountains, arid Central Asia[J]. Ecological Indicators, 2021, 126: 107702.

Zhang R, Wei W, Shang H, et al. A tree ring-based record of annual mass balance changes for the TS. Tuyuksuyskiy Glacier and its linkages to climate change in the Tianshan Mountains[J]. Quaternary Science Reviews, 2019, 205: 10-21.

Zhang T, Lu B, Zhang R, et al. A 256‑year‑long precipitation reconstruction for northern Kyrgyzstan based on tree‑ring width[J]. International Journal of Climatology, 2020, 40(3): 1477-1491.

Zhang T, Zhang R, Yuan Y, et al. Reconstructed precipitation on a centennial timescale from tree rings in the western Tien Shan Mountains, Central Asia[J]. Quaternary International, 2015, 358: 58-67.